# Pequi (*Caryocar brasiliense* Cambess)-Loaded Nanoemulsion, Orally Delivered, Modulates Inflammation in LPS-Induced Acute Lung Injury in Mice

**DOI:** 10.3390/pharmaceutics12111075

**Published:** 2020-11-11

**Authors:** Diego de Sá Coutinho, Jader Pires, Hyago Gomes, Adriana Raffin Pohlmann, Sílvia Stanisçuaski Guterres, Patrícia Machado Rodrigues e Silva, Marco Aurelio Martins, Stela Regina Ferrarini, Andressa Bernardi

**Affiliations:** 1Laboratory of Inflammation, Oswaldo Cruz Institute, Oswaldo Cruz Foundation, Rio de Janeiro 21040-360, Brazil; diego.dsc@ioc.fiocruz.br (D.d.S.C.); hyago.gomes@ioc.fiocruz.br (H.G.); patmar@ioc.fiocruz.br (P.M.R.eS.); marco.martins@fiocruz.br (M.A.M.); 2Institute of Health Sciences, Federal University of Mato Grosso, Sinop 78550-728, Brazil; jader129999@estudante.ufmt.br; 3Department of Organic Chemistry, Institute of Chemistry, Federal University of Rio Grande do Sul, Porto Alegre 91501-970, Brazil; adriana.pohlmann@ufrgs.br; 4College of Pharmacy, Federal University of Rio Grande do Sul, Porto Alegre 90610-000, Brazil; silvia.guterres@ufrgs.br

**Keywords:** acute lung injury (ALI)/acute respiratory distress syndrome (ARDS), nanoemulsion, drug delivery, pequi oil, oleic acid

## Abstract

Pequi is a Brazilian fruit used in folk medicine for pulmonary diseases treatment, but its oil presents bioavailability limitations. The use of nanocarriers can overcome this limitation. We developed nanoemulsions containing pequi oil (pequi-NE) and evaluated their effects in a lipopolysaccharide (LPS)-induced lung injury model. Free pequi oil or pequi-NE (20 mg/kg) was orally administered to A/J mice 16 and 4 h prior to intranasal LPS exposure, and the analyses were performed 24 h after LPS provocation. The physicochemical results revealed that pequi-NE comprised particles with mean diameter of 174–223 nm, low polydispersity index (0.11 ± 0.01), zeta potential of −7.13 ± 0.08 mV, and pH of 5.83 ± 0.12. In vivo evaluation showed that free pequi oil pretreatment reduced the influx of inflammatory cells into bronchoalveolar fluid (BALF), while pequi-NE completely abolished leukocyte accumulation. Moreover, pequi-NE, but not free pequi oil, reduced myeloperoxidase (MPO), TNF-α, IL-1β, IL-6, MCP-1, and KC levels. Similar anti-inflammatory effects were observed when LPS-exposed animals were pre-treated with the nanoemulsion containing pequi or oleic acid. These results suggest that the use of nanoemulsions as carriers enhances the anti-inflammatory properties of oleic acid-containing pequi oil. Moreover, pequi’s beneficial effect is likely due its high levels of oleic acid.

## 1. Introduction

Acute lung injury (ALI) and its most severe form, acute respiratory distress syndrome (ARDS), are inflammatory pulmonary disorders that result from various pathological insults, such as trauma, pneumonia, sepsis, endotoxemia, or multiple transfusions [1]. Despite the pathophysiology of ALI/ARDS being complex and not completely understood, it is known that neutrophils play a central role in these conditions by the release of granules and induction of oxidative injury, damaging the alveolar–capillary barrier [1,2]. The high mortality rate (approximately 40%) in critical care units, even with the latest advances in treatment, encourages new efforts to identify a more effective pharmacological approach for ALI/ARDS therapy [1].

Folk medicine, as an alternative therapy, has dramatically increased over the past three decades, with approximately 80% of the world’s population using these products as part of primary health care [3]. Brazil contains approximately 18% of all plant biodiversity worldwide, consisting of a rich source of natural products for phytotherapy [4]. The *Caryocar brasiliense* Cambess, popularly known as pequi or piqui, is a native fruit cultivated and consumed primarily in the Brazilian Cerrado [5]. Pequi plays an important role in the local economy due its use in the cosmetics industry and in traditional culinary methods [6]. Some studies have shown that pequi pulp contains several bioactive components with important anti-inflammatory, antioxidant, and healing properties, with oleic acid being the most predominant fatty acid [7,8]. Pequi pulp is also widely used in regional folk medicine to treat respiratory diseases, such as influenza, asthma, bronchitis, and infections [6]. Reports demonstrate that pretreatment with pequi seeds improved respiratory mechanics and decreased lung parenchyma damage of Wistar mice in a short-term secondhand-smoke exposure model [9]. In addition, supplementation with pequi oil or its extract resulted in antioxidant and anti-DNA damage properties in mice exposed to urethane-induced oxidative lung damage [10].

Pequi’s hydrophobic profile is a limitation of its therapeutic use. Therefore, nanotechnology formulations may represent an appropriate strategy to overcome this obstacle. Drug delivery based on nanotechnology has emerged as an important tool for therapy and disease prevention in biomedical applications [11]. Nanostructures used for drug delivery increase drug efficiency and safety, overcome the compound’s pharmacokinetic and pharmacodynamic limitations, improve bioavailability, enhance drug stability, optimize the dose of the drug, and reduce side effects. According to material and preparation methods, it is possible to obtain several nanoparticle structures, including nanocapsules, nanospheres, and nanoemulsions [12]. Nanoemulsions represent a nano-delivery system composed of a mixture of water and oil stabilized with surfactants, which, after spontaneous emulsification and organic solvent elimination, form oil in water emulsions. Nanoemulsions offer advantages, such as toxicological safety, low-cost fabrication, and the capacity to load large amounts of oil to protect them from evaporation, hydrolysis, and degradation [13,14].

Therefore, the aim of this study was to evaluate, for the first time, the effect of pretreatment with pequi oil in its free form or in a nanoemulsion system on pathological changes induced by lipopolysaccharide (LPS) exposure in mice. Since pequi oil pulp contains high amounts of oleic acid with well-characterized anti-inflammatory properties [15], this study also evaluated the implications of oleic acid in the beneficial effects of pequi.

## 2. Materials and Methods

### 2.1. Pequi Oil (Caryocar brasiliense Cambess) Extraction and Characterization

Pequi is a fruit that contains seeds coated with thorns and pulp. Mature fruits were collected in the city of Sinop (Mato Grosso, Brazil) and the internal cores were used to obtain the oil. The lumps were dried in an oven with forced ventilation at 40 °C for 72 h and comminuted in a knife mill (Wiley, Solab, Piracicaba, Brazil), an equipment used for grinding dried fibrous samples or samples containing water, fat, and/or oil. This equipment contains mobile and fixed knives that finely cut the sample. Decreasing the size to facilitate the organic extraction process, the oil was extracted with an organic solvent, hexane, at a ratio of 1:5 (*m/v*) in ultrasound for 2 h at a mean temperature of 40 °C. Subsequently, the extract was filtered to separate the solid waste and was then subjected to rotary evaporation at the same temperature to completely remove the solvent from the sample [8]. Pequi oil was characterized in triplicate according to the Brazilian Pharmacopeia V and by 1RMN (Bruker Avance spectrometer 400 MHz, Billerica, USA), and the unsaturated fatty acid composition, as oleic acid, was measured by gas chromatography by Agilent Technologies, GC: 7890A and MS: 5975C, Saint Clara, USA). Organoleptic and physicochemical characteristics, such as density, acidity, and pH, were assessed. The pequi oil was viscous and yellow-orange in color, presenting 0.76% humidity and 0.85% (*w/w*) acidity, indicated by the percentage of oleic acid, the primary compound of the oil. The relative density was equal to 0.887, and the oleic acid content in the oil was evaluated by 1RMN and compared to quantitative analysis by CG-MS, showing results of 24.40% and 25.90%, respectively. The percentage composition of oleic acid was approximately 65.50% compared to palmitic acid [11]. Access to genetic heritage and associated traditional knowledge of this manuscript was registered under CGen and SisGen portals under register number A0D4149.

### 2.2. Preparation of Nanoemulsion Formulations

Nanoemulsion formulations containing pequi oil were prepared by spontaneous emulsification as described by Bouchemal et al. [16] with slight modifications. The components of the organic phase (monostearate of sorbitan (40 mg), caprylic/capric triglycerides (150 mg), and pequi oil (10 mg) or oleic acid (2.59 mg)) were dissolved in acetone (27 mL) under magnetic stirring at 40 °C and injected into the aqueous phase (polysorbate 80 (76 mg) and water (53 mL)) under stirring, remaining in this condition for 10 min. After nanoemulsion formation, the solvents were eliminated in a rotative evaporator at 37 °C (R-200, Buchi, Flawil, Switzerland) and the formulation was concentrated to a final volume of 10 mL. The oleic acid concentration of the formulations was the same as in the pequi oil (25.9% *w/w*). The formulations obtained containing pequi oil and oleic acid were named pequi-NE and oleic acid-NE, respectively. For comparison, a nanoemulsion containing caprylic/capric triglycerides as oil, named blank nanoemulsion (BNE), was prepared without either pequi oil or oleic acid. All formulations were kept at room temperature, protected from light for 30 days.

### 2.3. Physicochemical Characterization of the Formulations

For each formulation batch, particle size distribution was assessed using Laser Diffraction Analysis (LD) (Mastersizer 2000, Worcestershire, UK) and the polydispersity (SPAN) was calculated. In the determination of droplet size by laser diffraction, the formulations were evaluated for an average equivalent sphere diameter (*d*_4.3_) and droplet size distribution (SPAN). SPAN was mathematically calculated by the equipment, being an important parameter for evaluating the polydispersity. The value of the SPAN was obtained according to the Equation: SPAN = (*d*_0.9_-*d*_0.1_)/*d*_0.5_, where *d*_0.9_, *d*_0.5_, and *d*_0.1_ are the diameters cumulative in the volumes of 90%, 50%, and 10% of the total population. The polydispersity index was determined by photon correlation spectroscopy, or Dynamic Light Scattering (DLS) (Zetasizer^®^ nano-ZS ZEN 3600, Worcestershire, UK) after dilution (1:500 *v/v*) of the samples with purified water. The method of cumulants was used to determine the hydrodynamic mean diameter (z-average diameter) and polydispersity index (PDI). These analyzes were performed on days 0, 15, and 30 after the development of the nanoemulsions. Zeta potential was determined using the same instrument after diluting the samples in 10 mmol/L NaCl aqueous solution. The pH was determined measured immediately after preparation of the formulations using a potentiometer (Highmed^®^, São Paulo, Brazil).

The morphology of the nanoemulsions were analyzed by a transmission electron microscope (TEM JEO), operating at the Electronic Microscopy Center of the Federal University of Rio Grande do Sul, using 120 kV. Therefore, pequi-NE and oleic acid-NE were diluted in ultrapure water and placed on the grids (formvar-carbon support film, electron microscopy sciences). Uranyl acetate 2% (*m/v*) was used with negative contrast.

### 2.4. Animals

Male A/J mice (18–20 g) were obtained from the Laboratory Animal Breeding Center of the Oswaldo Cruz Foundation. Animals were maintained in animal housing facilities with 5 animals per cage with a 12 h/12 h light/dark cycle at 25–28 °C with free access to food and water. All procedures followed the “Principles of Laboratory Animal Care” from the US National Institutes of Health and were approved by the Animal Ethics Commission of the Oswaldo Cruz Institution (CEUA IOC—License Number L-006/2016).

### 2.5. Lipopolysaccharide(LPS)-Induced Pulmonary Disorder Model and Treatment Protocol

As previously described in De Oliveira et al. [17], male A/J mice were anaesthetized with isoflurane (Cristália^®^, São Paulo, Brazil) aerosol with a constant flow of O_2_ and then underwent an intranasal stimulation with a solution containing LPS (25 µg in 25 µL sterile saline) (Sigma–Aldrich, St. Louis, MO, USA). The control group received only saline (VicMed, Rio de Janeiro, Brazil) intranasal administration. Treatment with 20 mg/kg pequi oil, pequi oil-loaded nanoemulsion (pequi-NE), oleic acid-loaded nanoemulsion (oleic acid-NE), or blank nanoemulsion (BNE) were performed orally 18 and 4 h before intranasal LPS exposure. Analysis were performed 24 h after LPS exposure.

### 2.6. Assessment of Pulmonary Function and Airway Hyper-Reactivity (AHR)

For respiratory mechanics testing, pulmonary elastance was evaluated using a whole-body plethysmography system (Buxco Electronics, Sharon, CT, USA) as previously described [18]. Animals were anaesthetized with 60 mg/kg of pentobarbital (i.p.) (Sigma–Aldrich, St. Louis, MO, USA), curarized with pancuronium bromide (Pavulon^®^) (Cristália^®^, São Paulo, Brazil), 1 mg/kg) (Sigma–Aldrich, St. Louis, MO, USA), tracheostomized, and connected to BUXCO equipment to be mechanically ventilated, and pulmonary function parameters were obtained. For AHR evaluation, increasing and cumulative concentrations of methacholine (3–27 mg/mL) were aerosolized.

### 2.7. Inflammatory Cell Analysis in the Airway Lumen

After AHR evaluation, mice were killed by anesthetic overdose of thiopental sodium (Cristalia^®^, São Paulo, Brazil), and bronchoalveolar lavage was performed as previously described [18]. Briefly, using a polyethylene cannula inserted into the animal’s trachea, a PBS solution (Sigma–Aldrich, St. Louis, MO, USA) containing EDTA (Sigma–Aldrich, St. Louis, MO, USA) was administered. The recovered lavage fluid was centrifuged, and cell pellets were resuspended for total leukocyte count by means of a Neubauer chamber. For cell count differentiation, cytospin slides were prepared from bronchoalveolar fluid (BALF) and stained by the May–Grunwald–Giemsa method.

### 2.8. Quantification of Myeloperoxidase (MPO) in Pulmonary Tissue

As previously described [17], lung tissue fragments were homogenized in Hank’s solution (Sigma–Aldrich, St. Louis, MO, USA), centrifuged, and pellets were resuspended in hypotonic solution followed by hypertonic NaCl solution for centrifugation again. Pellets were resuspended in hexadecyltrimethylammonium bromide (HTAB) (Sigma–Aldrich, St. Louis, MO, USA) and recentrifuged. Subsequently, 50 μL sample, 50 μL HTAB, and 50 μL ortho dianisidine were pipetted into a 96-well plate (Sigma–Aldrich, St. Louis, MO, USA). The plate was maintained at 37 °C for 15 min and then 50 μL H_2_O_2_ (Vetec, Rio de Janeiro, Brazil) was added to each well. After 10 min, sodium azide (1%) (Sigma–Aldrich, St. Louis, MO, USA) was added, and analysis was performed in a spectrophotometer at 460 nm wavelength. Results were adjusted by the amount of protein collected per lung.

### 2.9. Quantification of Inflammatory Mediators

Cytokine levels were quantified by ELISA in lung tissue samples homogenized in PBS containing a cocktail of protease inhibitors (Complete^®^,) (Roche Diagnostics, Mannheim, Germany) and centrifuged at 4 °C for 15 min at 10,000 *g*. Levels of MCP-1, TNF-α, IL-6, IL-1β, and KC were quantified in the supernatants using commercial kits according to the manufacturer’s instructions (R&D System, Minneapolis, MN, USA).

### 2.10. Oxidative Stress Analysis

Frozen lung fragments were homogenized in 500 μL potassium phosphate + EDTA buffer (KPE) at pH 7.5 and then centrifuged at 600 *g* for 10 min (4 °C). Supernatants were used for analysis of catalase enzyme activity and malondialdehyde (MDA) levels.

Catalase enzyme activity was measured by a method that employs the conversion of hydrogen peroxide (H_2_O_2_) to H_2_O and O_2_. An aliquot of 1 µL lung homogenates was added to 99 μL substrate mixture. The substrate mixture contained 0.3 mL hydrogen peroxide in 50 mL of 0.05 M phosphate buffer (pH 7.0). Initial and final absorbance was recorded by a spectrophotometer at 240 nm after 0, 30, and 60 s. Data are expressed as units of catalase per milligram of protein.

Pulmonary MDA levels induced by lipid peroxidation were determined using the thiobarbituric acid reactive substances (TBARS) method as previously described [17]. Lung tissue homogenates (100 μL) were mixed in 100 μL of 10% trichloroacetic acid and centrifuged for 15 min at 3600 *g* at 4 °C. Then, 150 µL of supernatant was collected and added to 150 μL thiobarbituric acid. Samples were heated at 95 °C for 10 min, and the reaction was stopped by placing samples on ice. MDA levels were determined by a spectrophotometer with absorbance at 532 nm. Data are expressed as nanomoles of MDA per milligram of protein.

### 2.11. Statistical Analysis

Statistical analysis was performed using the Prism package in GraphPad Software (version 5.0, San Diego, CA, USA). Data are expressed as the mean ± standard deviation of the mean (SD). Tests were performed using one-way ANOVA followed by the Newman–Keuls–Student test or two-way ANOVA followed by the Bonferroni test. *p* ≤ 0.05 was considered significant.

## 3. Results

### 3.1. Physicochemical Characterization of the Nanoemulsion Containing Pequi Oil

Nanoemulsion formulations BNE and oleic acid-NE showed a white opalescent color, and pequi-NE was opalescent, and slightly yellow. All formulations exhibited a macroscopically homogeneous appearance, without precipitate formation or phase separation. Table 1 shows the physicochemical parameters obtained for the nanoemulsions (BNE, pequi-NE, and oleic acid-NE) including volume-weighted mean diameter by volume (D [4, 3]), polydispersity (SPAN), z-average diameter by dynamic light scattering, and polydispersity index (PDI). Regarding polydispersity, SPAN and PDI indicated the quality of the formulation, demonstrate the homogeneity, and amplitude of the droplet distribution. The greater the size variation in the distribution, the greater the values. However, the SPAN and PDI calculations were different, based on the techniques of laser diffraction (LD) and droplet Brownian motion, respectively. According to the LD analysis, the formulations showed a mean ± standard deviation droplet size between 0.157 ± 0.01 of 0.174 ± 0.01 µm and SPAN around 1.51 ± 0.03. In addition, the DLS analysis demonstrated z-average diameters between 223 ± 0.02 and 179 ± 0.08 nm, with a low polydispersity index. The results showed no significant difference in the average diameter of the drops of BNE, pequi-NE, oleic acid-NE in the period of 30 days, remaining with 187 ± 0.02, 230 ± 0.01, and 155 ± 0.00, respectively. The zeta potential of nanoparticles varied close to −7 mV for BNE and pequi-NE. Oleic acid-NE presented a zeta potential more negative (Table 1) due to the carboxylic acid groups positioned at the droplet–water interface. Furthermore, formulations presented similar pH values, between 5.4 and 5.8. Results demonstrated that the nanoemulsion formulation containing pequi oil and oleic acid exhibited parameters suitable for application in biological models.

The nanoemulsions oleic acid-NE and pequi-NE were analyzed by transmission electron microscopy operating at 120 kV (Figure 1). TEM allowed us to analyze the morphology and shape of nanoemulsions. The droplets presented submicrometric diameters around 200 nm, and this result reinforced the results detailed in Table 1.

### 3.2. The Effect of Pequi Oil or Pequi-Loaded Nanoemulsion Treatment on LPS-Induced Pulmonary Inflammation

To compare the effect of free pequi oil and pequi-loaded nanoemulsion treatment, we used a short-term murine model of acute lung injury. Intranasal LPS administration resulted in increased leukocyte cell numbers in the mice alveolar space (Figure 2A), primarily composed of neutrophils (Figure 2B). Pretreatment with unloaded nanoemulsion failed to modify the total leukocyte and neutrophils counts in mice BALF. Oral administration of free pequi oil significantly reduced the migration of total leukocytes and neutrophils into the lung. Interestingly, treatment with pequi-NE at the same dose completely abolished the increased accumulation of these inflammatory cells (Figure 2A,B).

Moreover, treatment with BNE and pequi oil did not alter LPS-induced increases in MPO, a neutrophilic marker, in mouse lung tissue, while treatment with the nanoemulsion containing pequi oil dramatically attenuated the increase in MPO activity compared to the LPS group (Figure 3).

### 3.3. The Effect of Pequi Oil or Pequi-Loaded Nanoemulsion Treatment on LPS-Induced Pro-Inflammatory Cytokine Production

Measurement of inflammatory mediators showed that LPS provocation resulted in increased pulmonary levels of TNF-α (Figure 4A), IL-1β (Figure 4B), IL-6 (Figure 4C), MCP-1 (Figure 4D), and KC (Figure 4E). Oral treatment with pequi-NE reduced the levels of these cytokines in lung tissue, whereas treatment with the free pequi oil or BNE vehicle did not reduce levels of these inflammatory mediators.

### 3.4. Effect of Pequi Oil or Pequi-Loaded Nanoemulsion Treatment on LPS-Induced Airway Hyper-Reactivity

As shown in Figure 5, concentrations of methacholine aerosolization (3–27 mg/mL) resulted in a significant increase in lung elastance in the LPS exposed group compared to the saline control group. Treatment with 20 mg/kg of pequi oil in free form or loaded in nanoemulsion similarly inhibited the AHR, with no significant difference between them. Animals treated with BNE showed no significant difference when compared to animals challenged with LPS.

### 3.5. The Effect of Pequi-Loaded Nanoemulsion or Oleic Acid-Loaded Nanoemulsion Treatment on LPS-Induced Airway Hyper-Reactivity and Pulmonary Inflammation

To determine whether the effects of pequi-NE were closely related to its major component, we compared the effect of nanoemulsions containing pequi to nanoemulsion containing oleic acid in mouse lung inflammation and AHR induced by LPS exposure. As shown in Figure 6, pequi-NE (20 mg/kg) and oleic acid-NE (5 mg/kg) treatment equally decreased total leukocyte (Figure 6A) and neutrophil (Figure 6B) influx into the alveolar lumen, in addition to reducing MPO levels in lung tissue (Figure 7) compared to the group exposed to LPS and treated with BNE vehicle. Similarly, both treatments equally attenuated the enhanced LPS-induced pulmonary elastance followed by methacholine exposure (Figure 8).

### 3.6. Effect of Pequi-Loaded Nanoemulsion or Oleic Acid-Loaded Nanoemulsion Treatment on Pulmonary Oxidative Markers Induced by LPS Exposure

To assess the effect of nanoemulsions containing pequi or oleic acid on LPS-induced oxidative stress, we evaluated pulmonary catalase enzyme activity and MDA levels induced by LPS provocation in A/J mice. LPS stimulation resulted in decreased catalase antioxidant enzyme activity compared to the saline group. Treatment with pequi-NE and oleic acid-NE restored catalase activity to saline control levels (Figure 9A). Moreover, LPS induced a significant increase in MDA levels compared to the saline group. Treatment with pequi-NE and oleic acid-NE significantly decreased MDA levels compared to the LPS group (Figure 9B). No significant difference was found in catalase activity or MDA level in animals treated with BNE, if compared to the LPS untreated group.

## 4. Discussion

*Caryocar brasiliense* Cambess, known as pequi, is a native Brazilian fruit used in folk medicine to treat respiratory diseases [19] that presents protective activity in some rodent inflammatory models [20,21]. In this study, we investigated the effect of pequi oil, in free form or loaded in nanoemulsions, against pulmonary ALI/ARDS pathological changes triggered by intranasal LPS exposure in mice. We demonstrated for the first time that treatment with pequi-NE inhibited endotoxin-induced AHR and neutrophilic inflammatory influx into the airway lumen and lung tissue, reducing levels of crucial pro-inflammatory cytokines. We observed also that anti-inflammatory pequi activity can be correlated with the presence of its major oil constituent, oleic acid.

Current clinical applications of therapies are limited due to barriers, such as renal system filtering, first pass metabolism, premature removal by phagocytes, tortuous transport through the blood stream, or by drug lipophilicity. The use of nanocarriers has become a very common approach to avoid these issues [22]. Nanoemulsions were prepared by spontaneous emulsification [17]. This method is widely used due to its reproducibility and easy handling, resulting in an opalescent solution that can be directly used after preparation [23]. Nanoemulsions had a white bluish color due to the Tyndall effect characteristic of concentrated colloidal solutions. Pequi-NE presented a milky yellow opalescent color with a slight characteristic odor of the fruits used. All formulations were macroscopically homogeneous with a bluish reflection, resulting from the Brownian movement of the emulsion droplets. The techniques used to evaluate the diameters of the formulations showed the presence of nanometric droplets, without any microscopic sample. Besides that, the low polydispersity values demonstrated narrow size distributions and uniformity in the average diameters for all developed nanoemulsions. There was no significant difference (p.0.05) in the average diameter of the nanoemulsions in the evaluated period (30 days), indicating stability of the samples. The slightly acid pH values were consistent with the composition of the nanosystems in this study. It is noteworthy to mention that those values correspond to the pH of the aqueous phase, not related to the pH at the droplet–water interface. The acid–base balance of –COOH in contact with water produced carboxylate groups (–COO^−^), increasing the surface charge/surface potential at the interface. All these parameters qualified pequi-NE and oleic acid-NE formulations for application in biological models. Values of the zeta potential remained between −6.72 and −13.76, adequate to avoid aggregation of the droplets since they were prepared with a nonionic surfactant, such as polysorbate 80, with stereo hindrance mechanisms [24].

To analyze whether the incorporation of pequi oil in the nanoemulsion resulted in an increase in its pharmacological effects, we evaluated the impact of its administration in mice submitted to an experimental model of ALI. Experimental models using LPS are characterized for being rapid once instillation results in lung inflammation, which occurs 4–48 h after exposure and is reproducible and reliable for assaying the biological effects of new drugs, being a standard model for inducing experimental ALI and ARDS [1]. Previous work has shown that LPS instillation results in an influx of leukocytes, such as neutrophils and macrophages, into the lungs of mice [18]. During the acute stage of inflammation, neutrophils migrate to inflammatory sites, releasing proteases that degrade the extracellular matrix components, which increases the inflammatory response; or releasing cytokines that favor the accumulation of more leukocytes [25]. In contrast, alveolar macrophages play an important role in the development of lung injury through their exacerbated secretion of mediators, such as oxidants and proteinases [26]. We observed that oral pretreatment with pequi-NE, but not with pequi oil in free form, reduced the influx of neutrophils and macrophages collected in the BALF. Our results are consistent with studies reporting that pequi administration reduces neutrophil and monocytes present in blood in a model of age-related inflammation in mice [5]. Moreover, pretreatment with pequi reduced leukocyte infiltration, primarily polymorphonuclear cells, in the joint cavity in a model of zymosan-induced arthritis in rat knee joint [27]. Of note, BALF leukocyte reduction observed in our study was accompanied by attenuated levels of pro-inflammatory cytokines and MPO levels in lung tissue. MPO is the most abundant protein present in intracellular neutrophil granules, [28]. It is already well described in the literature that LPS exposure induces increased MPO activity in mice [17]. In accordance with our results, de Figueiredo et al. observed that neutrophils obtained from human blood donors incubated with pequi oil exhibited reduced MPO release [29].

Interestingly, the anti-inflammatory effect observed in response to the nanoemulsion containing pequi administration was not reproduced by treatment with the compound in its free form, suggesting that the nanoparticle formulation resulted in an increase in bioavailability and vectorization to inflammatory sites. This enhanced bioavailability may result from protection of drugs against destructive elements and increased absorption into the gastrointestinal tract [30]. Several mechanisms have been described involving gastrointestinal absorption of nanoparticles, such as increased mucus adhesion, transport by cellular channels, capture by intestinal epithelial cells, and capture by lymph nodes in the ileum [31]. Regarding the passive vectorization, it is based on the combination of high permeability and improved retention phenomena that occurs in inflamed tissues. Increased permeability observed in the vasculature of inflamed tissues results from unregulated angiogenesis and increased expression and activation of vascular permeability factors, which facilitate the entrance of nanoparticles. Additionally, there is poor drainage, resulting in nanoparticle retention within the inflamed tissues [32,33].

Studies in experimental models of ALI/ARDS demonstrate that an increased number of leukocytes and pro-inflammatory mediator generation in the lungs of mice is generally accompanied by a reduced lung function and airway hyper-reactivity response [17]. Elastance is defined as the pressure required to inflate the lungs and can be a useful parameter to measure airway hyper-reactivity in ALI/ARDS conditions [34]. We observed that treatment with the nanoemulsion containing pequi oil reduced AHR in mice stimulated with LPS and exposed to methacholine. Curiously, we observed that despite not affecting the inflammatory response, administration of pequi oil in its free form attenuated the increased pulmonary elastance in mice. The influx of inflammatory cells is often associated with airway hyper-responsiveness, suggesting a direct relationship between these parameters. This conclusion, however, is controversial because anti-inflammatory treatment with inhaled steroids, in some circumstances, does not inhibit AHR [35]. Previous studies have indicated that AHR induced by systemic LPS administration is not related to the presence of neutrophils or their mediators in the pulmonary microvasculature [36]. It is possible that the mechanism leading to airway hyper-reactivity in our experimental model cannot be directly related to the presence of leukocytes in BALF and lung tissue. Thus, we can suggest that the dose used for free pequi oil treatment in our study was satisfactory for reducing the AHR but not enough for reducing the massive inflammatory response.

It has been reported that pequi’s biological properties are closely related to its oil composition. The pequi structure is composed of an epicarp, an external mesocarp, a pulp that is rich in oil, and a rigid endocarp with spines and white kernel (seed) [37]. Information on pequi oil pulp chemical composition is important for adequate exploitation of this fruit as a therapeutic agent. Several studies have elucidated pequi composition, and in all of them, oleic acid was the predominant fatty acid [7,37,38]. Oleic acid, a ω-9 monounsaturated fatty acid, is the primary constituent of olive oil commonly consumed in a Mediterranean diet and associated with good health [15]. For this reason, we compared the effect of administering nanoemulsions containing oleic acid with a formulation containing pequi in our ALI experimental protocol. We observed that treatment with the nanoemulsion containing pequi or containing the equivalent amount of oleic acid presented similar results, reducing AHR and the inflammatory parameters that were analyzed. However, oleic acid activity in the immune system is controversial, and some studies have shown that oleic acid can induce lung injury due to lipid embolism. This can be explained because this fatty acid can suppress, enhance, or synergize neutrophil function, depending on the experimental condition and the applied dose [39,40]. Moreover, the studies show also that the use of nanocarriers can enhance the oleic acid anti-inflammatory activity [41]. In another study, it was observed that nanostructured lipids containing oleic acid inhibit human neutrophil superoxide generation and elastase release induced by albumin in vitro, and ameliorate the neutrophil infiltration and severity of mice skin inflammation induced by leukotriene B4 in vivo [42]. These studies are agreement with our results, showing that the use of nanocarriers can enhance the beneficial effect of oleic acid administration.

It is known that pequi medicinal use is directly related with its rich natural antioxidant composition, including oleic acid, and represents an alternative therapy for oxidative stress conditions [38,43]. Oxidative stress is an imbalance between the production of reactive oxygen species (ROS) and antioxidant defense mechanisms, sometimes leading to tissue damage [44]. Excessive ROS generated by the injured endothelium/epithelium and leukocytes plays a crucial role in ALI/ARDS progression, amplifying the pulmonary damage and edema, while impairing gas exchange. Catalase is an antioxidant enzyme that converts hydrogen peroxide (H_2_O_2_) into water and oxygen, aiming to protect the organism from oxidant-induced damage [45]. We observed in our study that the ALI/ARDS experimental model induced by LPS exposure resulted in reduced antioxidant enzyme catalase activity and increased lipid peroxidation damage in mouse lung tissue. Interestingly, pequi and oleic acid-NE treatment reversed pulmonary oxidative stress. Our results are according to studies showing promising pequi activity with respect to the inhibition of pre-formed free radicals in vitro analyzed by the 2,2-diphenyl-1-picryl-hydrazyl-hydrate free radical scavenging method (DPPH assay) [5,19]. Moreover, pequi reduced lipid peroxidation in an in vivo aluminum-induced neurotoxicity and neuroinflammation model [46], in an exhaustive exercise-induced liver inflammation model [43], in liver in an atherosclerosis model [47], and in an urethane-induced lung tumor model [10].

## 5. Conclusions

The findings reported herein suggest that treatment with nanoemulsions containing pequi oil exerts anti-inflammatory and antioxidant effects in an ALI/ARDS induced by LPS in mice. This effect is mediated, at least in part, by the presence of oleic acid, the majority compound of the oil. The most powerful evidence generated from this report shows that the nanostructuration strategy proved to be crucial to improve both the effect of pequi oil as well as of oleic acid administered orally. Taken together, our findings suggest that nanoemulsions containing pequi oil or containing oleic acid represent an alternative pharmacological strategy for the treatment of pulmonary inflammatory disorders.

## Figures and Tables

**Figure 1 pharmaceutics-12-01075-f001:**
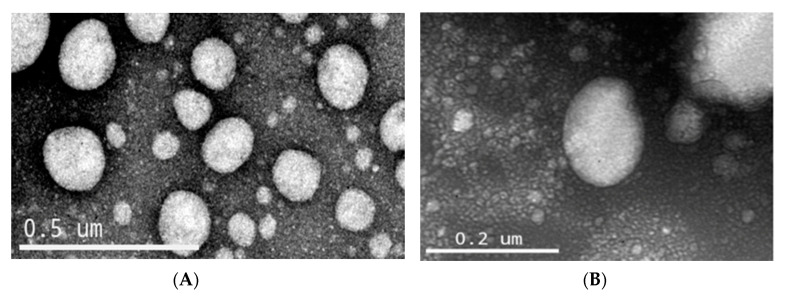
Photomicrography of nanoemulsions. (**A**) Oleic acid-NE and (**B**) pequi-NE.

**Figure 2 pharmaceutics-12-01075-f002:**
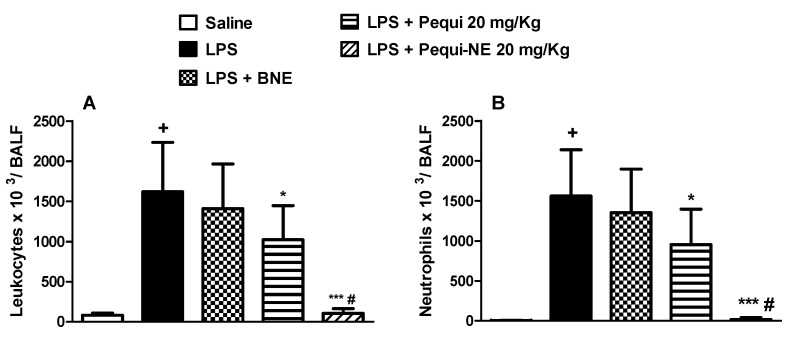
Effect of oral treatment with pequi or pequi-loaded nanoemulsion on total cells (**A**) and neutrophils (**B**) influx to mice alveolar space induced by lipopolysaccharide (LPS) instillation. Animals were orally pre-treated with 20 mg/kg of pequi oil, pequi oil-loaded nanoemulsion (pequi-NE), or blank nanoemulsion (BNE) 16 and 4 h before intranasal LPS (25 μg/25 μL) exposure. After 24 h of LPS instillation, cells were collected from bronchoalveolar lavage fluid (BALF) for cell counts. Data are expressed as the mean ± SD from 8 animals. + *p* < 0.05 compared to the saline group; * *p* < 0.05 and *** *p* < 0.001 compared to the LPS group; # *p* < 0.05 compared to the pequi group.

**Figure 3 pharmaceutics-12-01075-f003:**
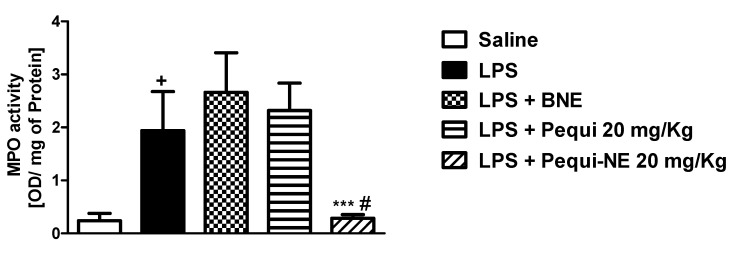
Effect of oral treatment with pequi or pequi-loaded nanoemulsion on increased pulmonary myeloperoxidase (MPO) levels induced by LPS instillation in mice. Animals were orally pre-treated with 20 mg/kg of pequi oil, pequi oil-loaded nanoemulsion (pequi-NE), or blank nanoemulsion (BNE) 16 and 4 h before intranasal LPS (25 μg/25 μL) exposure. After 24 h of LPS instillation, the lungs were perfused and collected to assess myeloperoxidase (MPO) activity. The optical density (OD) was adjusted by the amount of protein per lung. Data are expressed as the mean ± SD from 7–8 animals. + *p* < 0.05 compared to the saline group; *** *p* < 0.001 compared to the LPS group; # *p* < 0.05 compared to the pequi group.

**Figure 4 pharmaceutics-12-01075-f004:**
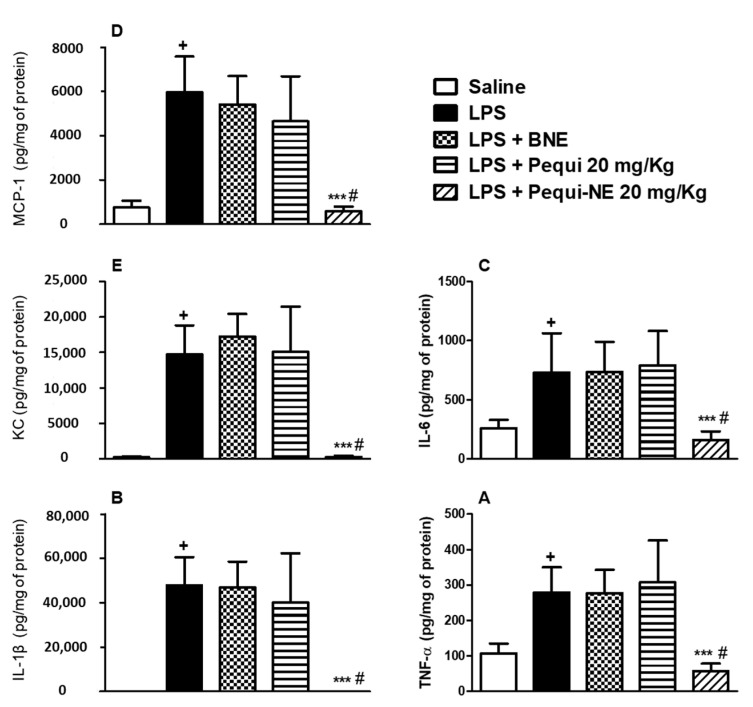
Effect of oral treatment with pequi or pequi-loaded nanoemulsion on enhanced pulmonary cytokines levels induced by LPS instillation in mice. Animals were orally pre-treated with 20 mg/kg of pequi oil, pequi oil-loaded nanoemulsion (pequi-NE), or blank nanoemulsion (BNE) 16 and 4 h before intranasal LPS (25 μg/25 μL) exposure. After 24 h of LPS instillation, lungs were perfused and collected to measure levels of tumor necrosis factor alpha (TNF-α) (**A**), interleukin (IL)-1β (**B**), IL-6 (**C**), monocyte chemoattractant protein-1 (MCP-1) (**D**), and keratinocyte-derived chemokine (KC) (**E**), data are expressed as the mean ± SD from 5–7 animals. + *p* < 0.05 compared to the saline group; *** *p* < 0.001 compared to the LPS group; # *p* < 0.05 compared to the pequi group.

**Figure 5 pharmaceutics-12-01075-f005:**
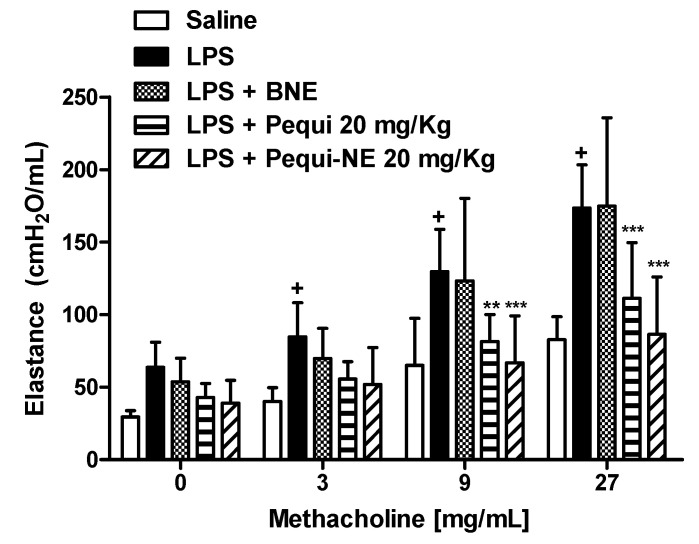
Effect of oral treatment with pequi or pequi-loaded nanoemulsions on mice airway hyper-reactivity induced by LPS instillation. Animals were orally pre-treated with 20 mg/kg pequi oil, pequi oil-loaded nanoemulsion (pequi-NE), or blank nanoemulsion (BNE) 16 and 4 h before intranasal LPS (25 μg/25 μL) exposure. Airway responses were measured as changes in lung elastance induced by concentrations of methacholine (3–27 mg/mL) 24 h after LPS instillation. Data are expressed as the mean ± SD from 7–8 animals. + *p* < 0.05 compared to the saline group; ** *p* < 0.01 and *** *p* < 0.001 compared to the LPS group.

**Figure 6 pharmaceutics-12-01075-f006:**
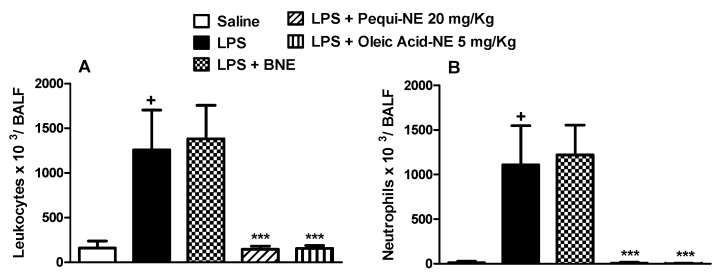
Effect of oral treatment with pequi or oleic acid-loaded nanoemulsion on total cells (**A**) and neutrophils (**B**) influx to mice alveolar space induced by LPS instillation. Animals were orally pre-treated with 20 mg/kg of pequi-loaded nanoemulsion (pequi-NE), with 5 mg/kg of oleic acid-loaded nanoemulsion (oleic acid-NE), or with blank nanoemulsion (BNE) 16 and 4 h before intranasal LPS (25 μg/25 μL) exposure. After 24 h of LPS instillation, cells were collected from bronchoalveolar lavage fluid (BALF) for cells count. Data are expressed as the mean ± SD from 7–8 animals. + *p* < 0.05 compared to the saline group; *** *p* < 0.001 compared to the LPS group.

**Figure 7 pharmaceutics-12-01075-f007:**
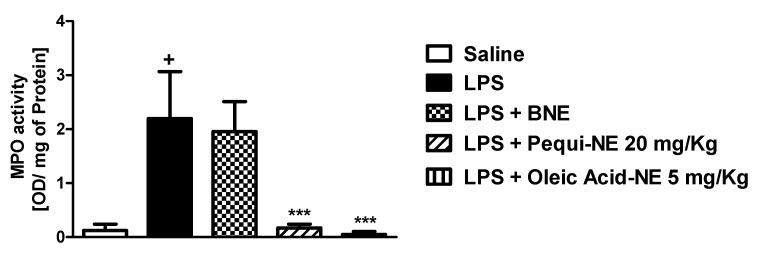
Effect of oral treatment with pequi or oleic acid-loaded nanoemulsion on increased pulmonary MPO levels induced by LPS instillation in mice. Animals were orally pre-treated with 20 mg/kg of pequi-loaded nanoemulsion (pequi-NE), with 5 mg/kg of oleic acid-loaded nanoemulsion (oleic acid-NE) or with blank nanoemulsion (BNE) 16 and 4 h before intranasal LPS (25 μg/25 μL) exposure. After 24 h of LPS instillation, the lungs were perfused and collected to assess myeloperoxidase (MPO) activity. The optical density (OD) was adjusted by the amount of protein per lung. Data are expressed as the mean ± SD from 7–8 animals. + *p* < 0.05 compared to the saline group; *** *p* < 0.001 compared to the LPS group.

**Figure 8 pharmaceutics-12-01075-f008:**
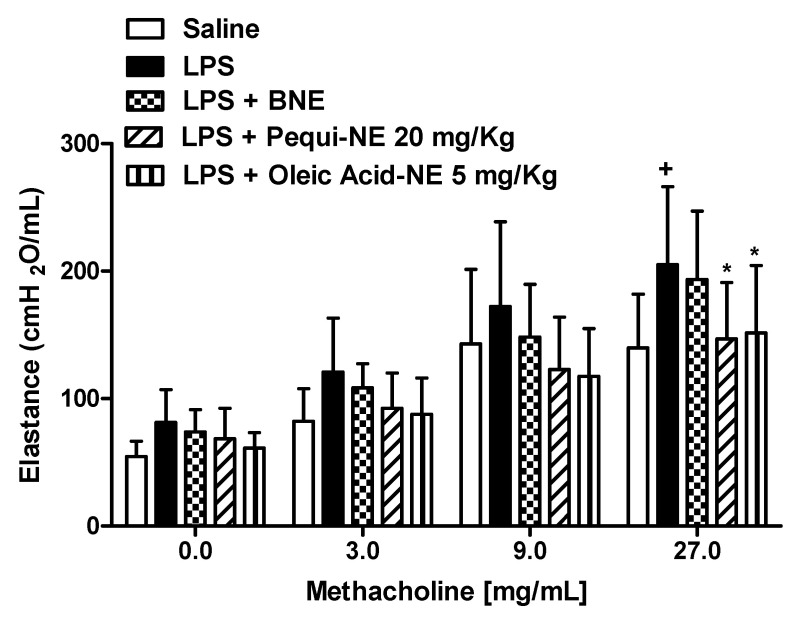
Effect of oral treatment with pequi or oleic acid-loaded nanoemulsion on mice airway hyper-reactivity induced by LPS instillation. Animals were orally pre-treated with 20 mg/kg of pequi-loaded nanoemulsion (pequi-NE), with 5 mg/kg of oleic acid-loaded nanoemulsion (oleic acid-NE), or with blank nanoemulsion (BNE) 16 and 4 h before intranasal LPS (25 μg/25 μL) exposure. Airway responses were measured as changes in lung elastance induced by concentrations of methacholine (3–27 mg/mL) 24 h after LPS instillation. Data are expressed as the mean ± SD from 7–8 animals. + *p* < 0.05 compared to the saline group; * *p* < 0.05 compared to the LPS group.

**Figure 9 pharmaceutics-12-01075-f009:**
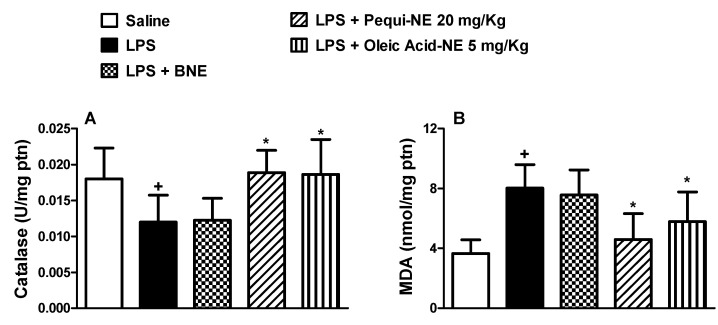
Effect of oral treatment with pequi or oleic acid-loaded nanoemulsion on mice pulmonary oxidative imbalance induced by LPS exposure. Animals were orally pre-treated with 20 mg/kg of pequi-loaded nanoemulsion (pequi-NE), with 5 mg/kg of oleic acid-loaded nanoemulsion (oleic acid-NE) or with blank nanoemulsion (BNE) 16 and 4 h before intranasal LPS (25 μg/25 μL) exposure. After 24 h of LPS instillation, lungs were perfused and collected to measure catalase activity (**A**) and malondialdehyde (MDA) levels (**B**). Catalase activity was measured by a decrease in H_2_O_2_, while the thiobarbituric acid reactive substances (TBARS) method was used to analyze MDA products as an index of oxidative damage induced by lipid peroxidation. Data are expressed as the mean ± SD from 6–8 animals. + *p* < 0.05 compared to the saline group; * *p* < 0.05 compared to the LPS group.

**Table 1 pharmaceutics-12-01075-t001:** Physicochemical analysis of BNE, pequi-NE, and oleic Acid-NE.

Formulations				
		BNE ^5^	Pequi-NE ^6^	Oleic Acid-NE ^7^
Laser Diffraction (LD) ^1^	Mean diameter (nm)	157 ± 0.01	174 ± 0.01	163 ± 0.02
	SPAN ^2^	1.13 ± 0.01	1.51 ± 0.03	1.60 ± 0.02
Dynamic Light Scattering (DLS) ^3^	Z-average diameter (nm)	186 ± 0.01	223 ± 0.02	179 ± 0.08
	PDI ^4^	0.09 ± 0.02	0.11 ± 0.01	0.11 ± 0.01
Zeta Potential	Zeta potential (mV)	−6.72 ± 0.13	−7.13 ± 0.08	−13.6 ± 0.72
pH		5.44 ± 0.27	5.83 ± 0.12	5..51 ± 0.09

Data are expressed as mean ± SD of 3 batches. ^1^ LD = Laser Diffraction and ^2^ SPAN = Droplet size distribution; ^3^ DLS = Dynamic Light Scattering and ^4^ PDI = Polydispersity index; ^5^ BNE = Blank nanoemulsion (BNE); ^6^ Pequi-NE = Pequi-loaded nanoemulsion; ^7^ Oleic acid-NE = Oleic acid nanoemulsion.

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
