# Peer review of "Pequi (Caryocar brasiliense Cambess)-Loaded Nanoemulsion, Orally Delivered, Modulates Inflammation in LPS-Induced Acute Lung Injury in Mice"

_pharmaceutics, 2020, doi:10.3390/pharmaceutics12111075_

Round 1

Reviewer 1 Report

SIGNIFICANCE OF THE WORK:

The aims of this study are the development of nanoemulsions containing Pequi oil (Pequi-NE) and the evaluation of their effects in a lipopolysaccharide (LPS)-induced lung injury model.

METHODOLOGY:

The research is thorough, and the conclusions are supported by the experiments.

TEXT:

The manuscript is, in general, well-crafted, and written. Please correct the following typos:

P 3: 40ºC, 1RMN

P4: O2

P5: 4ºC (H2O2) H2O O2

P14, reduces, can induces, can enhances

COMMENTS:

In my opinion, this is a nice piece of work. I would like to make some comments to the author to enrich the manuscript:

I would suggest the authors to write the conclusions of their work. The discussion section is too long and complex. For instance, one of conclusions of the work that I consider interesting and it should be emphasized is the following statement: “These studies are agreement with our results showing that the use of nanocarriers can enhances the beneficial effect of oleic acid administration”

The authors claim that “we compared the effect of administering nanoemulsions containing oleic acid with a formulation containing Pequi in our ALI experimental protocol. We observed that treatment with the nanoemulsion containing Pequi or containing the equivalent amount of oleic acid presented similar results, reducing AHR and the inflammatory parameters that were analysed.” Do the authors consider that there is any advantage in using the Pequi oil compared with oleic acid alone?

The authors claim that their “nanoemulsions containing Pequi exerts anti-inflammatory and antioxidant effect”. Have the authors quantified this activity? Which is their reference?

Reviewer 2 Report

1- The title of the manuscript is unexpressive to the work done, so it has to be re-constructed to include the route of administration.

2- The abstract not cover the main aspects of the work.

3- What are the limitations of Pequi oil mentioned in the abstract?

4- The manuscript language is very poor and it has to be revised for grammatical and language errors.

5- The introduction part is not sufficiently expressive and need to be rewritten focusing on the role of Pequi oil in the treatment of inflammatory lung diseases.

6- Under 2.2. Preparation of nano emulsion formulations.

a- Isolation of Pequi's oil from the Pequi fruits and its standardization is ambiguous and need more clarification.

b- What is meant by hammer/knife mill and its model and its operation conditions?

c- Why the author not undergo a preliminary study on a group of surfactants and cosurfactants?

7- Under 2.3. Physicochemical characterization of the formulations.

a- What is meant by the polydispersity (span) was calculated?

b- Why there the in vitro release and permeation studies has not been conducted on the prepared nano emulsion?

8- The results need deep discussions and correlations with previously published works.

Reviewer 3 Report

This manuscript deals with preparation, characterization and evaluation of pequi oil-loaded nanoemulsion for modulation of inflammation in LPS-induced acute lung injury in mice. This is an interesting piece of work and does hold the novelty for a possible publication in “Pharmaceutics”. However, the following moderate revision needs to be done before this paper could be accepted for publication.

  1. Abstract – please provide (a) the “low polydispersity index” value in parenthesis, (b) include the unit of zeta potential, and (c) last sentence should be deleted and instead “enhances the Pequi anti-inflammatory properties” can be replaced by “enhances the anti-inflammatory properties of oleic acid-containing Pequi oil”. “Mean ± standard deviation” should be removed.
  2. Abstract & Remaining text – the abbreviations “BAL” and “BALF” are interchangingly used for bronchoalveolar fluid.
  3. Keywords – Is it “ALI/SDRA” or “ALI/ARDS”?
  4. “Mean ± SEM” should be replaced with more common “Mean ± standard deviation” or “Mean ± SD” throughout the text as well as tables/figures and their captions.
  5. Section 2.2 – How was acetone eliminated?. Is it rotative or rotary? For comparison, the nanoemulsion containing caprylic/capric triglycerides as oil should be denoted as “Blank nanoemulsion (BNE)” (not as “drug unloaded nanoemulsion”) in this place and throughout the text including figures/tables and their captions.
  6. Section 2.3 – why the pequi-nanoemulsion was not characterized for TEM image? It is better to include a TEM image and a photograph of pequi-nanoemulsion
  7. Section 2.4 – please cite a reference for “Animal Ethics Commission”.
  8. Section 3.1 & Table 1 – “close to 7 mV” should be “close to -7 mV”. The mean diameters should be provided in “nm” and not “mm”. Please clearly distinguish between the particle size measurement by LD and DLS methods along with the associated PDI values both in the text and table 1 footnote.
  9. The nanoemulsions with “zeta potential” values not being >+30 or <-30 usually indicate their instability. How do the authors justify the stability of pequi-NE amid its low zeta potential value? What is “stereo hindrance mechanisms” mentioned under section 4?
  10. Figures 1 & 5 – the description of “total cells (A) and neutrophils (B) should be brought to the bold-faced main caption at the beginning and not in the footnote description of the caption.
  11. Figures 4 & 5 – please describe the “not significant” feature of last two treatments in the text, especially with their association with the dose “27 mg/mL methacholine”.
  12. Please describe the full form of all the abbreviations used in both Table 1 and all the figures (including in legends and x/y-axes) in the table footnote and figure captions, respectively, as each figure and table should be clearly understandable without reference to the text (stand-alone).
  13. Section numbers should be numbered sequentially. Please double-check the order and correct them appropriately.
  14. The notation of “h” and “hours” are interchangingly used in various places in the text and figure captions, which should be denoted as “h” in all places. Also “seconds” should be denoted as “s” and “ml” should be corrected as “mL” throughout the text as well as tables/figures and their captions.
  15. In the usage of H2O2, H2O, O2, the number should be appropriately subscripted.
  16. Typographical, grammar and wording errors should be double-checked throughout the text and corrected, especially “satisfactory for reduces….for reduces the massive” should be corrected as “satisfactory for reducing…for reducing the massive” and “nanocarriers can enchances” as “nanocarriers can enhance” in Page 14.
  17. The last paragraph of section 4 should be separated to be under the separate “Conclusion” heading.
  18. Considering the total length of the paper, the number of references are too high (73) and it should be reduced to at least two-thirds. This can be done (a) by removing the old references (39,45,47,60,61,64), (b) by cutting down the multiple reference citations in single location and (c) by choosing to cite references that can be used for two or more points of explanation.

Round 2

Reviewer 2 Report

Taking into consideration of the reviewers comments greatly improved the quality of the manuscript and make it suitable for publication.